# Protocol on a systematic review of nomenclature and outcomes in children with complex critical illness in Paediatric Critical Care: The basis for consensus definition

**Sofia Cuevas-Asturias**[1,2]*, **Claire Rafferty**[3], **Hannah Mitchell**[4], **William Tremlett**[5], **Padmanabhan Ramnarayan**[1,2‡], **Natalie Pattison**[1,6,7‡]

**1** Department of Surgery & Cancer, Imperial College London, London, United Kingdom, **2** Paediatric Intensive Care Unit, Imperial College Healthcare NHS Trust, London, United Kingdom, **3** Paediatric Intensive Care Unit, The Bristol Royal Hospital for Children, Bristol, United Kingdom, **4** Institute of Child Health, University College London, London, United Kingdom, **5** Paediatric Intensive Care Unit, Birmingham Children's Hospital, Birmingham, United Kingdom, **6** University of Hertfordshire, Hatfield, United Kingdom, **7** East and North Herts NHS Trust, Stevenage, England

‡ PR and NP are Joint Senior Authors.
* sofia.cuevas-asturias06@imperial.ac.uk

## Abstract

### Introduction

Paediatric Critical Care (PCC) supports the recovery of children with severe illness. In the UK, there are 30 PCC units with a total of approximately 400 beds. There is constant demand for these beds with a mean five-day length of stay and admissions increasing at a greater rate than age-specific population growth. Prolonged stay patients account for approximately half of all PCC patient bed days. Children with complex critical illness (CCI) need input from multiple different teams alongside support for their family. CCI often become prolonged PCC-stay patients too. Internationally, there is variation in the definition of CCI, this creates service variation and tensions around what resources can be provided including discharge planning, provision, and support. **Objective:** The face of Paediatric Critical Care, in the UK and internationally has changed over the last ten years with a growing cohort of complex critically ill patients. This systematic review aims to look at current nomenclature, criteria, and outcome measures of priority in this undefined patient population.

### Methods and materials

**Inclusion criteria:** All types of studies examining children with complex critical illness (age <18 years) admitted to any paediatric critical care. The review is registered on Prospero. Medline, Embase, Maternity and Infant care, The Cochrane Library, the Cumulative Index to Nursing and Allied Health Literature (CINAHL), and the Trip database will be searched from 2014 to May 2024. The search was limited to ten years as children with complex critical illness are a relatively new concept within PCC. Therefore, the timeline

**Data availability statement:** No datasets were generated or analysed during the current study. All relevant data from this study will be made available upon study completion.

**Funding:** The author(s) received no specific funding for this work.

**Competing interests:** The authors have declared that no competing interests exist.

was limited to increase the accuracy and applicability of the review. Search limits included all languages, excluded the setting of neonatal intensive care, and age>18 years old. The final search strategy was developed in Medline and peer-reviewed by a health research librarian not involved in the study. This was translated to other databases as appropriate. Four independent reviewers will screen citations for eligible studies and perform data extraction.

## Discussion

A systematic review methodology has been used to develop a broad understanding of the literature which will be used to develop further work in this area. Using a rigorous and stepwise approach, the whole spectrum of scientific publications on children with complex critical illness in paediatric intensive care will be reviewed, ensuring this study is as comprehensive as possible. This includes quantitative, qualitative, theoretical, and grey literature. A limitation of this systematic review is the use of many terms to describe children with complex critical illness in the literature resulting in a high number of publications on this topic.

## Introduction

### Rationale

Paediatric Critical Care (PCC) has 30 units across the United Kingdom, comprising a total of approximately 400 beds [1]. The occupancy rate of PCC beds runs critically high (above 85% occupancy) most of the year and studies have shown an increasing demand for PCC beds with concern that demand may soon outstrip resources [1]. One reason for increasing demand is the greater number of a heterogeneous group of children with complex critical illness (CCI) who often have prolonged lengths of stay (PLOS). The definition for a PLOS PCC admission varies from >14 to >28 days. Over the last two decades, PLOS PCC admissions have increased significantly and now account for between 42-51% of PCC patient days [2]. PLOS PCC patients have a high resource utilisation and a median overall hospital length of stay of 98 days [3]. PCC has a decreasing overall trend in mortality, but PLOS patients have significantly higher mortality than the general PCC population [3].

Existing classification systems of CCI patients are variable and do not allow for accurate documentation of the incidence of PCC admission and subsequent mortality or morbidity [4].

**Children with complex critical illness (CCI).** Children with CCI comprise a diverse population group, both within the United Kingdom and around the world. They do not possess a unified definition or comprehensive criteria within PCC. Children with CCI currently have a variety of definitions in the literature, for example, some definitions refer to children with pre-existing complex needs [5], multi-morbidity [6] or technological dependence [7] that require PCC support. Another concept of children with CCI is a previously well child who whilst in PCC becomes a child with CCI due to the evolving multi-morbidity or evolving chronic technology-dependence. A survey conducted in the U.K. in 2024 with the Paediatric Critical Care Society Study Group (PCCS-SG) looked at provisions for children with CCI. This showed variable patient identification, management, and a paucity of standardised care pathways [8]. As a result, the economic impact, mortality, and morbidity of this patient group is unknown.

In 2022, Edwards et al published a concise review that focused on "severe chronic illness" in PCC and proposed a range of strategies to meet the unique needs of these patients within PCC and beyond [4]. It touched upon some of the definitions and terminology for children with CCI. This review aims to delve deeper into the terminology and associated criteria.

Following this, Zorko et al conducted a scoping review in January 2023 that specifically looked at the term "chronic critical illness" within PCC [9]. This review included studies that provided a specific definition for chronic critical illness or prolonged stay but did not encompass medical complexity at admission, repeated admissions, or technological dependence. Additionally, Zorko's review did not provide an overview of all current terminology and associated criteria for children with complex critical illness.

In the context of children with CCI, the development of a single all-encompassing definition presents a significant challenge. Alternatively, an umbrella term may emerge to encapsulate the diverse range of conditions and symptoms that these children may experience. The challenge lies in devising a definition to capture the complexity and evolving nature of their illnesses while maintaining clarity and precision.

Children with CCI present unique challenges to PCC:

- (Frequently) prolonged length of stay

- Complex medical regimens

- Complicated family dynamics

- Multiple specialist and allied healthcare professional input amongst others [8].

## Objectives

This systematic review aims to primarily examine the current nomenclature and associated criteria of children with complex critical illness. Additionally, the systematic review will investigate secondary outcome measures of priority in this undefined patient population.

# Materials and methods

## Study design

This systematic review protocol was developed using systematic review published methodology [2,10–14]. It is reported according to the Preferred Reporting Items for Systematic Review and Meta-analyses (PRISMA-P) [10], this is detailed in the Checklist in S1 Checklist. The study was registered in the International Prospective Register of Systematic Reviews (PROSPERO; registration number is: CRD42024529649) after confirming no similar systematic reviews were already registered. An accurate audit trail has been kept allowing for reproducibility [13]. This study is exempt from ethics approval as this work is carried out on published documents.

## Eligibility criteria

The following criteria will be considered for this review

**Types of participants.**  Studies looking at critically ill children (age < 18 years) admitted to any paediatric critical care unit (PCC) globally, identified with the following terms:

- Paediatric complex critical illness

- Complex chronic conditions

- Prolonged or long-stay PCC admission

- Medical Complexity

- Severe or chronic critical illness

- Severe neurologic impairment

- Technology-dependent children

These chosen search terms were based on concepts identified in the Edwards et al., 2022 review [4].

**Types of interventions.**  All types of interventions will be considered. All types of outcome measures will be included. We will group outcomes into themes to analyse research areas of interest to date.

## Information sources

This review is reviewing experimental studies, randomised controlled trials, non-randomised controlled trials, before and after studies, and interrupted time-series studies. In addition, analytical observational studies including prospective and retrospective cohort studies, case-control studies, and analytical cross-sectional studies are being considered for inclusion. This review will consider descriptive observational study designs and descriptive cross-sectional studies for inclusion. Qualitative studies are also being considered that focus on qualitative data including, but not limited to, designs such as grounded theory, ethnography, qualitative description, and action research. Children with CCI are a heterogeneous group with a wide range of clinical management needs within PCC. To gain a deeper understanding of the topic and present a balanced perspective, "Tier 1" grey literature will be reviewed. This includes book chapters, government reports, think tank publications, and policy documents according to Adams et al. [15,16].

**Exclusions.**  The study evaluates only the adult population or evaluates adults and paediatric populations but does not report separate data for each population. This study will exclude editorials, commentaries/opinion pieces, abstract/conference proceedings, case series, and individual case reports. Additionally, systematic reviews that meet the inclusion criteria will be excluded but have their reference list of reviews searched for relevant citations. As children with complex critical illness are an emerging concept in the field of PCC, literature published prior to 2014 will be excluded unless cited by a systematic review.

## Search strategy

The search strategy includes all languages, using translation online tools [17] and language capabilities of author team (English, Spanish and Portuguese) and full-text papers. The search strategy aimed to locate both published and unpublished studies. An initial limited search of Ovid MEDLINE was done to identify articles on the topic. The text words contained in the titles and abstracts of relevant articles, and the index terms used to describe the articles were used to develop a full search strategy for The Cochrane Library, the Cumulative Index to Nursing and Allied Health Literature (CINAHL), MEDLINE, EMBASE, Trip database (see Search strategy in S1 Appendix).

The search strategy, including all identified keywords and index terms, will be adapted for each included database and/or information source. The reference list of all included sources of evidence will be screened for additional studies.

Ongoing consultation with a health librarian will aid the specificity and comprehensiveness of the search. The MEDLINE search strategy is within the PROSPERO registration appendix to allow another reviewer the ability to replicate and/or evaluate the search. The impact of

novel treatments on enhancing survival has grown more prevalent over the past ten years, as evidenced by a four-fold increase in long-stay patients in PCC, which is associated with a higher mortality rate [18]. Data management and selection process

Following the search, all identified citations will be collated and uploaded into Covidence, and duplicates removed. Titles and abstracts will be screened by three independent reviewers for assessment against the inclusion criteria. All sampling decisions will be transparent and justified with a PRISMA search flow diagram [2].

The full text of selected citations will be assessed in detail against the inclusion criteria by two or more independent reviewers. Reasons for the exclusion of sources of evidence at full-text review that do not meet the inclusion criteria will be recorded and reported in the systematic review. The search results and study inclusion process will be reported in full in the final systematic review and presented in a Preferred Reporting Items for Systematic Reviews and Meta-analyses (PRISMA) flow diagram [19].

## Data collection process

Data will be extracted from papers included in the review by three independent reviewers using the systematic review software tool Covidence. The data extracted will include specific details about the participants, concept, context, study methods, and key findings relevant to this systematic review.

Data will also be clustered geographically into related subgroups to identify themes whilst maintaining a detailed audit trail. A narrative summary will accompany the tabulated results and will describe how the results relate to the systematic review objectives.

A draft extraction form is provided in S1 Table. The draft data extraction tool will be modified as required during each data extraction process. Modifications will be detailed in the systematic review. Any disagreements that arise between the reviewers will be resolved through discussion, or with an additional reviewer/s. If appropriate, authors of papers will be contacted to request missing or additional data, where required. Critical appraisal of individual sources of evidence will be done.

## Quality assessment

The quality and validity of the studies selected will be assessed using the critical appraisal skills program checklists (CASP) to mitigate bias [20–23]. Eight studies (four quantitative and four qualitative studies) will be randomly picked to test and further refine the quality assessment tool by all reviewers. Consistencies and inconsistencies between reviewers will be noted, alongside this, the scoring system will be modified according to problems encountered.

Qualitative studies will be evaluated utilising the GRADE-CERQual approach [24]. The evidence will be presented in a summary of qualitative findings table, incorporating CERQual assessment of confidence in the evidence scores for each summary of review finding.

Quantitative data will be geographically clustered into related subgroups to identify themes while maintaining a detailed audit trail. A narrative summary will accompany the tabulated results and will describe how the results relate to the review objectives and questions.

## Outcomes

The main outcome of the research will provide an evidence base to inform and influence the development of a definition for paediatric complex critical illness. As there is no current standard for this definition, this review will evaluate how medical complexity, chronic critical illness, and prolonged PCC admissions have been defined. We will also look at how the definition was developed and/or validated by primary study.

To address the secondary objective of this study we aim to review all outcome measures within the literature including:

- Health-related outcomes using validated tools where possible (e.g., functional status, severity of illness, co-morbidities, quality of life, symptom burden, unmet needs, satisfaction, rates of hospitalisation).

- Process outcomes such as quality of care, family-centre care, professional education initiatives, cost evaluations and resource utilisation.

Interventions will be pooled to look at common attributes and potential benefits/shortfalls. Overall, the findings of this review will be used to inform a future program of research aimed at improving the identification, management, and outcomes of paediatric complex critical illness patients.

## Patient and public involvement

This protocol was developed with public and patient involvement. They were involved from the first principal onwards. The research questions have been developed as a result of the author teams' engagement with children with complex critical illness and their families in PCC. A parent of a child with CCI has commented on the methods and was integral to the protocol finalisation. A parent advisory group is being formed for the consensus work following the review.

## Ethics

This proposal seeks to conduct a systematic review that will not involve human or animal subjects. Therefore, review by an ethics committee is not required and patient consent for publication is not applicable.

## Discussion

The systematic review methodology possesses several strengths including the utilisation of the PRISMA-P guidelines which helps ensure a reproducible, rigorous, and stepwise approach. The whole spectrum of scientific publications on children with complex critical illness in paediatric intensive care is planned to be reviewed with a published search strategy, ensuring this study is as comprehensive as possible. This includes quantitative, qualitative, theoretical, and grey literature. With the search strategy developed with health librarian input. Throughout the process, we aim to have two reviewers screening and extracting the data with conflicts resolved by an additional reviewer, strengthening the methodology.

A limitation of this review is the use of many terms to describe complex critical patients in the literature, this will likely result in a large heterogeneity of data requiring multiple types of CASP checklists. The literature will be used to form a UK-specific consensus definition for CCI. An international solution is also required, but with the large variation in models of PCC provision, the author team feel a UK-specific definition will be a starting point for the future development of international guidelines.

As part of the next stages of this work, we are forming a national parent advisory group. This will form part of the patient and public involvement and engagement (PPIE) strategy to aid in dissemination through patient and public networks within complex critical patients. We aim to publish the results in academic literature and presentations at conferences in the United Kingdom and internationally to ensure findings are internationally available to practitioners and researchers.

The new knowledge produced may contribute to the education and training of undergraduate and postgraduate paediatric intensive care multidisciplinary health professionals and allied health and social care professionals.

Overall, the review plans to build on international existing scoping and concise reviews to provide up-to-date evidence for the United Kingdom [4,9]. It also looks at a wider scope of definitions in comparison to Zorko's work and includes a wider range of publications to give the best evidence base for a consensus definition development for children with complex critical illness in PCC.

## Supporting information

**S1 Checklist. PRISMA-P (Preferred Reporting Items for Systematic review and Meta-Analysis Protocols) 2015 checklist.**
(DOCX)

**S1 Appendix. Search strategy for MEDLINE.**
(DOCX)

**S1 Table. Data extraction form** .
(DOCX)

## Author contributions

**Conceptualization:** Sofia Cuevas-Asturias, Padmanabhan Ramnarayan.

**Data curation:** Sofia Cuevas-Asturias.

**Formal analysis:** Sofia Cuevas-Asturias.

**Investigation:** Sofia Cuevas-Asturias.

**Methodology:** Sofia Cuevas-Asturias, Padmanabhan Ramnarayan.

**Project administration:** Sofia Cuevas-Asturias.

**Resources:** Sofia Cuevas-Asturias.

**Supervision:** Natalie Pattison, Padmanabhan Ramnarayan.

**Validation:** Natalie Pattison.

**Visualization:** Natalie Pattison.

**Writing – original draft:** Sofia Cuevas-Asturias, Hannah Mitchell.

**Writing – review & editing:** Sofia Cuevas-Asturias, Claire Rafferty, Hannah Mitchell, William Tremlett, Padmanabhan Ramnarayan.

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
