## [Decision Letter · Decision Letter 0]

16 Jul 2024

PONE-D-24-25160Protocol on a Systematic review on nomenclature and outcomes in children with complex critical illness in Paediatric Critical Care - The basis for consensus definition.PLOS ONE

Dear Dr. Cuevas-Asturias,

Thank you for submitting your manuscript to PLOS ONE. After careful consideration, we feel that it has merit but does not fully meet PLOS ONE’s publication criteria as it currently stands. Therefore, we invite you to submit a revised version of the manuscript that addresses the points raised during the review process. Please submit your revised manuscript by Aug 30 2024 11:59PM. If you will need more time than this to complete your revisions, please reply to this message or contact the journal office at plosone@plos.org . Please include the following items when submitting your revised manuscript:

We look forward to receiving your revised manuscript.

Kind regards,

Tai-Heng Chen, M.D., Ph.D.

Academic Editor

PLOS ONE

Journal Requirements:

Reviewers' comments:

Reviewer's Responses to Questions

**Comments to the Author**

1. Does the manuscript provide a valid rationale for the proposed study, with clearly identified and justified research questions?

Reviewer #1: Yes

Reviewer #2: Partly

2. Is the protocol technically sound and planned in a manner that will lead to a meaningful outcome and allow testing the stated hypotheses?

Reviewer #1: Yes

Reviewer #2: Partly

3. Is the methodology feasible and described in sufficient detail to allow the work to be replicable?

Reviewer #1: Yes

Reviewer #2: Yes

4. Have the authors described where all data underlying the findings will be made available when the study is complete?

Reviewer #1: Yes

Reviewer #2: Yes

5. Is the manuscript presented in an intelligible fashion and written in standard English?

Reviewer #1: Yes

Reviewer #2: Yes

6. Review Comments to the Author

You may also provide optional suggestions and comments to authors that they might find helpful in planning their study.

Reviewer #1: This is an important topic, and the authors should be commended for tackling what will no doubt be a large piece of work reviewing relevant papers.

I have only a few minor comments:

In the abstract, the authors say "nationally" - as the journal has international readership, this should be specified "in the UK"

It may be helpful for the authors to include some idea of what makes a "complex critical illness" - I realise that the aim of the overall work is to find a unifying definition of this, but some background on the groups the authors aim to include would be helpful, eg whether they are predominantly referring to children with underlying complex needs who happen to deteriorate and require ICU support vs children previously well children with an acute illness which is complex on the basis of number of systems involved or multiple coexisting pathologies. This is elaborated on in the "eligibilty" section but some more introduction would be useful and would make it clearer to the reader what they are expecting to read about.

The authors plan to limit their search to papers published in the last decade - what is the rationale for this decision?

Will the search be limited to English language, full text papers? Will conference abstracts be included? This information should be included in the search strategy.

Reviewer #2: Thank you for the opportunity to review this manuscript, which details a systematic review protocol on nomenclature and outcomes in children with complex critical illness in the pediatric ICU.

While this topic is of great importance considering the increasing medical complexity of children admitted to the pediatric ICU, the manuscript is not well organized. For a better organizational structure, I suggest combining PLOS One requirements for main sections (Introduction, Materials & Methods, Discussion) with the PRISMA-P criteria as subsections within each main section. The use of future tense throughout the manuscript is also at times confusing, because it seems that the search strategy has already been formed (Appendix 1) and the trial has already been registered, but these types of activities are stated to occur in the future. Together these two issues make it difficult to clearly understand the review methods.

Please see the attached PDF for additional specific suggestions to improve clarity and quality for publication.

7. PLOS authors have the option to publish the peer review history of their article (what does this mean? ). If published, this will include your full peer review and any attached files.

**Do you want your identity to be public for this peer review?** For information about this choice, including consent withdrawal, please see our Privacy Policy .

Reviewer #1: **Yes: ** Amanda Friend

Reviewer #2: No

---

## [Author Response · Author response to Decision Letter 1]

9 Aug 2024

Thank you to both reviewers for their feedback and time. All the suggested revisions have been reviewed and the changes are detailed alongside a tracked-changes manuscript in the attached files. Many thanks for your time.

---

## [Decision Letter · Decision Letter 1]

4 Sep 2024

PONE-D-24-25160R1Protocol on a systematic review of nomenclature and outcomes in children with complex critical illness in Paediatric Intensive Care: The basis for consensus definitionPLOS ONE

Dear Dr. Cuevas-Asturias,

Thank you for submitting your manuscript to PLOS ONE. After careful consideration, we feel that it has merit but does not fully meet PLOS ONE’s publication criteria as it currently stands. Therefore, we invite you to submit a revised version of the manuscript that addresses the points raised during the review process.

We look forward to receiving your revised manuscript.

Kind regards,

Tai-Heng Chen, M.D., Ph.D.

Academic Editor

PLOS ONE

Journal Requirements:

Reviewers' comments:

Reviewer's Responses to Questions

**Comments to the Author**

1. Does the manuscript provide a valid rationale for the proposed study, with clearly identified and justified research questions?

Reviewer #1: Yes

Reviewer #2: Yes

2. Is the protocol technically sound and planned in a manner that will lead to a meaningful outcome and allow testing the stated hypotheses?

Reviewer #1: Yes

Reviewer #2: Partly

3. Is the methodology feasible and described in sufficient detail to allow the work to be replicable?

Reviewer #1: Yes

Reviewer #2: No

4. Have the authors described where all data underlying the findings will be made available when the study is complete?

Reviewer #1: Yes

Reviewer #2: No

5. Is the manuscript presented in an intelligible fashion and written in standard English?

Reviewer #1: Yes

Reviewer #2: Yes

6. Review Comments to the Author

You may also provide optional suggestions and comments to authors that they might find helpful in planning their study.

Reviewer #1: Many thanks for submitting a revised abstract; this is now clearer and much easier to follow. I am happy to now recommend this for publication.

Reviewer #2: Thank you for the opportunity to review this revised manuscript, which details a systematic review protocol on nomenclature and outcomes in children with complex critical illness in the pediatric ICU.

I would like to thank the reviewers for their work in refocusing and reorganizing this manuscript. Additional suggestions to continue to clarify and refine the review methods are included in the attached document.

7. PLOS authors have the option to publish the peer review history of their article (what does this mean? ). If published, this will include your full peer review and any attached files.

**Do you want your identity to be public for this peer review?** For information about this choice, including consent withdrawal, please see our Privacy Policy .

Reviewer #1: No

Reviewer #2: No

---

## [Author Response · Author response to Decision Letter 2]

13 Dec 2024

Please see second revisions letter attached. Thank you for taking the time to review the manuscript and I hope the changes made are well received.

Thank you,

Sofia Cuevas-Asturias

---

## [Decision Letter · Decision Letter 2]

14 Jan 2025

Protocol on a Systematic Review of nomenclature and outcomes in children with complex critical illness in Paediatric Critical Care: The basis for consensus definition

PONE-D-24-25160R2

Dear Dr. Cuevas-Asturias,

We’re pleased to inform you that your manuscript has been judged scientifically suitable for publication and will be formally accepted for publication once it meets all outstanding technical requirements.

Kind regards,

Tai-Heng Chen, M.D., Ph.D.

Academic Editor

PLOS ONE

Reviewers' comments:

Reviewer's Responses to Questions

**Comments to the Author**

1. Does the manuscript provide a valid rationale for the proposed study, with clearly identified and justified research questions?

Reviewer #1: Yes

2. Is the protocol technically sound and planned in a manner that will lead to a meaningful outcome and allow testing the stated hypotheses?

Reviewer #1: Yes

3. Is the methodology feasible and described in sufficient detail to allow the work to be replicable?

Reviewer #1: Yes

4. Have the authors described where all data underlying the findings will be made available when the study is complete?

Reviewer #1: Yes

5. Is the manuscript presented in an intelligible fashion and written in standard English?

Reviewer #1: Yes

6. Review Comments to the Author

You may also provide optional suggestions and comments to authors that they might find helpful in planning their study.

Reviewer #1: Many thanks for submitting a revised manuscript; although I was happy to accept the previous submission, I note my fellow reviewer's points and appreciate the author's considerations.

7. PLOS authors have the option to publish the peer review history of their article (what does this mean? ). If published, this will include your full peer review and any attached files.

**Do you want your identity to be public for this peer review?** For information about this choice, including consent withdrawal, please see our Privacy Policy .

Reviewer #1: No

---

## [Editor Report · Acceptance letter]

PONE-D-24-25160R2

PLOS ONE

Dear Dr. Cuevas-Asturias,

I'm pleased to inform you that your manuscript has been deemed suitable for publication in PLOS ONE. Congratulations! Your manuscript is now being handed over to our production team.

Kind regards,

on behalf of

Dr. Tai-Heng Chen

Academic Editor

PLOS ONE